# Influence of the Number of Queens on Nest Establishment: Native and Invasive Ant Species

**DOI:** 10.3390/ani11030591

**Published:** 2021-02-24

**Authors:** Irene Castañeda, Elsa Bonnaud, Franck Courchamp, Gloria Luque

**Affiliations:** 1Community Ecology Group, UMR INRA 1202 BIOGECO, Université de Bordeaux 1, 33405 Talence, France; 2CNRS, AgroParisTech, Ecologie Systématique Evolution, Université Paris-Saclay, 91405 Orsay, France; elsa.bonnaud@universite-paris-saclay.fr (E.B.); franck.courchamp@u-psud.fr (F.C.); gmluque@advancedconservation.org (G.L.)

**Keywords:** invasive, ant, native, establishment, queens, workers, *Linepithema humile*, *Tapinoma nigerrimum*

## Abstract

**Simple Summary:**

Nest establishment is a critical stage of the ant life cycle because it determines the chances of colony success. Here, we study the effect of different numbers of queens (i.e., one or six) on the position of queens and workers inside and outside the artificial nests of an invasive (*Linepithema humile*) and Mediterranean native (*Tapinoma nigerrimum*) species. Our results suggest that queens in nests with six queens entered the nest faster than single queens. Similarly, during nest establishment, workers in nests with six queens entered the nest faster, with this effect being more pronounced for the native species. Once nests were established, fewer workers were engaged in outside-nest tasks in nests with six queens. This was especially true for workers of the native species engaged in patrolling. These results suggest that the number of queens can influence both queen and worker behavior, and that invasive and native species have different responses.

**Abstract:**

As a critical stage in the life cycle of ant colonies, nest establishment depends on external and internal factors. This study investigates the effect of the number of queens on queen and worker behavior during nest establishment in invasive Argentine ants (*Linepitema humile*) and native Mediterranean *Tapinoma nigerrimum*. We set up experimental colonies with the same number of workers but with one or six queens. At different time points, we recorded the positions of queens and workers inside and outside the nest. Our results highlight the influence of the number of queens on the position of queens and workers with between-species differences. Queens of both species entered the nests more quickly when there were six queens. During nest establishment, more workers were inside nests with six queens for both species, with this effect being greater for *T*. *nigerrimum*. Once nests were established, fewer workers of both species were engaged in nest maintenance and feeding in nests with six queens; *T*. *nigerrimum* had fewer workers engaged in patrolling. These results suggest that the number of queens is a key factor driving queen and worker behavior during and after nest establishment with different species responses.

## 1. Introduction

In ants, as well as most other species, the propagule size (i.e., the number of introduced individuals), propagule number (i.e., the number of introduction events), and propagule pressure (i.e., the composite measure between propagule size and number) determine the likelihood of establishment [1,2,3]. In theory, a larger initial propagule size and more frequent introduction events increase the chances of establishment success [3]. Despite the use of laboratory, field, or combined approaches [4,5,6] to study the role of propagule size and pressure in ant establishment success, they are among the most poorly understood factors of ant ecology.

In dependent colony foundation, the success of nest establishment is also influenced by the role played by each caste. For example, in the absence of workers, queens of polygynous species are unable to produce new workers and thus rarely survive. Moreover, brood production is limited by worker provision rather than the egg-laying capacity of queens [7]. Consequently, the number of workers and queens has a critical influence on the successful establishment of emerging ant colonies and the subsequent population dynamics [8]. Regardless of the organization level (i.e., individual or colony), experimental studies often address the effect of workers on a species’ competitive ability [9]. However, queens can also affect the productivity and survival of workers [10], as well as the behavior of the queen [11] and workers [12]. The most efficient management strategy to avoid the establishment of invasive or potentially invasive species is prevention [13,14]. Thus, determining the role of factors such as propagule size and number of queens in experimental nests can shed light on nest establishment strategies, particularly for invasive ant species.

In this study, we evaluated the effect of species and the number of queens on both queen and worker behavior for nest establishment in a laboratory experimental setup. This type of setup is the best option to compare species because we can control abiotic (e.g., temperature and humidity) and biotic (e.g., number of species interacting) factors. As we were interested in comparing two species in isolation, this study was not possible to conduct in the field, as many species are simultaneously present and all species interact with each other. We focused on two ant species, one invasive (*Linepithema humile*) and one native (*Tapinoma nigerrimum*). Among invasive ant species, *L*. *humile* is a cosmopolitan species that has successfully invaded all continents apart from Antarctica, and it continues to spread [15]. Its success is mainly due to its trophic habits (e.g., mass recruitment and omnivorous diet), reproduction strategy (i.e., high polygyny) and colony structure (e.g., colonies with lack of boundaries due to the absence of aggression). These characteristics allow *L*. *humile* to rapidly recruit workers and accumulate food, leading to the formation of super colonies [16]. For instance, in the Mediterranean basin, *L*. *humile* has formed a super colony that extends from southern Spain along the Mediterranean coast to Italy [17]. In this area, *L*. *humile* lives in sympatry with the native *T. nigerrimum*, which has similar life history traits. This Mediterranean species is abundant and widespread, forms polydomous and polygynous colonies, and exhibits mass recruitment to obtain food sources. Consequently, these species have similar space and trophic requirements which likely promotes competition for those resources, as has already been shown in experimental conditions [18]. Therefore, understanding what characteristics may provide advantages for one species over the other could provide insight into the potential value of the native species *T*. *nigerrimum* as a biological control agent of the invasive *L*. *humile*. Studies comparing these two species have been focused on differences in worker behavior and abundance [19,20]. Here, we focus on the potential effect of queens on nest establishment, one of the critical phases of species invasion [7].

## 2. Material and Methods

### 2.1. Ant Collection

In April 2014, we collected *L. humile* queens, workers, and brood from La Ciotat in southern France (43.18° N, 5.61° E). We simultaneously collected *T. nigerrimum* queens, workers, and brood from nearby Auriol in southern France (43.37° N, 5.64° E) where *L*. *humile* nests are also present. These field stations located near Marseille, France, have a Mediterranean climate with moderately low temperatures in winter and dry hot periods in summer. Vegetation types at both field stations are mainly composed of shrub species (*Retama* sp. and *Thymus* sp.) and forest (*Pinus* sp. and *Quercus* sp.). The site surroundings were dense residential areas and large expanses of agricultural land. We collected workers, queens, and brood at different stages of development for both species.

### 2.2. Experimental Design

Ant colonies were fed with a sucrose solution and a protein-based solution twice a week [21]. The inner surfaces of the large plastic boxes where the ants were kept were coated with fluoropolymer resin (fluon) to stop the ants from escaping. The laboratory temperature was kept at 24–26 °C throughout the experiment, which is optimal for both species [22].

Each experimental colony was composed of a nest (length: 15.3 cm, width: 7.8 cm, height: 1.5 cm) and a foraging arena (length: 15 cm, width: 15 cm, height: 15 cm) connected by a small transparent tube. Nests were mainly made of plaster and kept between two plates of transparent polyvinyl chloride (PVC). Nests were covered with transparent red paper that produced darkness for ants, thus simulating natural conditions. The foraging arena was a plastic box with a plaster bottom where we placed food.

We used an experimental design with two initial numbers of queens (1 and 6 queens) and 300 workers for each species. We named the colonies according to the four treatments: 1-queen invasive, 1-queen native, 6-queen native, and 6-queen invasive. We chose these numbers of queens and workers based on the results of [14], which showed that both workers and queens have a positive effect on the productivity of the other caste and that queens have a positive effect on worker survivorship. We set up the corresponding number of queens and workers for each treatment and made 20 replicates per treatment (*n* = 80 nests, with a total of 280 queens and 24,000 workers). These initial colonies did not contain brood.

For each experimental colony, we placed the workers and queen(s) inside the foraging arena and took photos to monitor queen and worker behavior. To maintain the humidity of the nests, we added 2 mL of water twice a week using a syringe. We also cleaned the foraging arenas by removing the mounds of dead individuals made by workers as well as any remaining food. Most workers and all queens stayed inside the nests while we performed the nest cleaning and maintenance.

To follow the queen and worker positions in the experimental setups at different times, we took photos (Panasonic LUMIX DMC-FZ45) of the experimental nest chambers and the corresponding foraging arena. We considered three main positions of workers or queens: (i) inside the nest; (ii) in the foraging arena (outside the nest around food and water supplies), and (iii) climbing the walls of the foraging arena (Appendix A).

During the first day after setup, we recorded the position of the ants at four different times after they were placed inside the experimental arena: 0.5 h, 1 h, 1.5 h, and 2 h. We also recorded the nests the following day (i.e., after 24 h) and 7 days after the start of the experimental setup. Photos were analyzed using the “counting” tool in the Adobe Photoshop software to determine the number of individuals in each position of the experimental nest (Appendix A).

According to the biology of ants, we fitted two models to consider two periods after setup: establishment of the colony (i.e., from setup to 2 h) and maintenance of the colony over a longer period (i.e., 24 h and 7 days after setup). As these models presented similar results to those obtained from the full model (i.e., six time points) (Appendix A), we have only presented and discussed results from the first two models.

### 2.3. Statistical Analyses

#### 2.3.1. Queens during Nest Establishment

We assessed the time for the first queen to enter the nest using Kaplan–Meier (K–M) survival analysis [23]. In our case, the outcome event is a queen situated inside the nest. We used K–M estimates with censored data, since queens from three nests had not yet entered the nest by the end of observation on the first day (i.e., the event of interest did not occur for those queens).

#### 2.3.2. Workers Inside the Nests

We fitted a generalized linear mixed model (GLMM) during nest establishment (i.e., between 0.5 h to 2 h after setup) and another model for 24 h and 7 days after setup, aiming to analyze the relationship between the number of workers inside the nests, the number of queens, and the species for each of these periods. We used the number of queens (as a factor), species, time, and their interaction as fixed effects. We included nest identity (ID) as the random effect, because there were multiple correlated observations from the same nests (i.e., number of workers inside the nest at four different times). Using nest ID as a random effect allowed us to consider the correlation between multiple observations from the same nest while only estimating one variance. We used a Poisson distribution, as the dependent variable (i.e., number of workers) is count data (i.e., real positive numbers), and a negative binomial distribution in the case of overdispersion. We used the deviance goodness of fit test and inspected the normality of residuals, heteroscedasticity, and independence of random effects (Appendix A). The goodness-of-fit test based on deviance is the log-likelihood ratio statistic for testing the fitted model against the saturated model in which there is a regression coefficient for every observation. When significant interactions were detected, pairwise comparisons were done with the package emmeans [24].
(i) Workers ~queens∗species∗time+(1| nest ID)

#### 2.3.3. Workers Outside the Nests

Workers can be engaged in different tasks outside the nest such as patrolling, nest maintenance, and feeding [25]. Once nests were established, we quantified the number of workers outside the nest engaged in one of the following tasks: patrolling (i.e., workers on walls), nest maintenance (i.e., workers in the foraging arena carrying plaster pieces and cleaning the nest), and feeding (i.e., workers located inside or around protein and/or sugar tubes). We used a generalized linear model (GLM) to analyze each of these tasks as follows: the number of workers as a response variable, and the number of queens (as a factor), species, and their interaction as fixed variables. We used the deviance goodness of fit test to select models and analysis of variance (ANOVA) to compare them (Appendix A).
(ii) Workers ~queens∗species

Statistical analyses were conducted using R (R Core Team 2019) with an α-level of 0.05. We used the lme4 package [26] to perform generalized linear mixed-effects analysis. We used the survival package [27] to perform survival analysis. *p*-values were obtained by likelihood ratio tests of the full model, with the effect compared to the model without the effect. The plots presented here were made using the ggplot2 package [28], and values are presented as mean ± 95% confidence interval (CI).

## 3. Results

### 3.1. Queens during Nest Establishment

There was a difference in the time spent by the first queen to enter the nest between the number of queens (likelihood ratio test = 21.60, df = 12, *p* < 0.001); queens from six-queens nests entered faster than queens from one-queen nests (Hazard Ratio: HR = 2.923, *p* < 0.001) (Appendix A, Figure 1). Nevertheless, there was no differences in the time spent by the first queen to enter the nest between species in one-queen nests (likelihood ratio test = 2.680, df = 1, *p* = 0.100) (Appendix A, Figure 1) nor six-queen nests (likelihood ratio test = 1.200, df = 1, *p* = 0.300) (Appendix A, Figure 1).

### 3.2. Workers Inside the Nest

We compared the negative binomial model with random intercept (model 1) with a model without random intercept (model 2). The model with random intercept was significantly better that the simpler model (chi square = 217.86, *p* < 0.001) and had smaller Akaike Information Criterion (AIC) (model 1: df = 10, AIC = 3278, vs. model 2: df = 9, AIC = 3494). Both models showed similar AIC (no interaction: df = 7, AIC = 3279, vs. model 1: df = 10, AIC = 3278) and there was not no significant difference between both models them (chi square = 7.312, *p* = 0.063), so we removed the time interaction. The model with species interaction had smaller AIC (model 1: df = 10, AIC = 3278 vs. no interaction: df = 6, AIC = 3285) and there was a significant difference between both models (chi square = 15.163, *p* = 0.004), so we kept the species interaction.

During nest establishment (from 0.5 h to 2 h after setup), there was a significant effect of the number of queens and the number of workers entering inside the nest; there were more workers inside the nest with six queens (z = 4.248, *p* < 0.001). *T*. *nigerrimun* had significantly more workers entering the nest at all times than *L*. *humile* (z = 5.441, *p* < 0.001), but the interaction between the species and the number of queens was also significant (z = −2.898, *p* = 0.004). The effect of time was also significant (z = 13.270, *p* < 0.001) because the number of workers entering the nest increased with time since most workers spend most of their time inside the nest with the queens (Figure 2, Appendix A). Pairwise comparisons between the number of queens and species combinations showed that the number of queens had a significant effect only for *L*. *humile* (Figure 2, Appendix A).

We compared the negative binomial model with random intercept (model 1) with a model without random intercept (model 2). Model 1 was significantly better than model 2 (chi square = 5.753, *p* = 0.016) and had smaller AIC (model 1: df = 10, AIC = 759, vs. model 2: df = 9, AIC = 763). The model with time interaction had smaller AIC (model 1: df = 10, AIC = 759 vs. no interaction: df = 7, AIC = 764) and there was a significant difference between both models (chi square = 10.775, *p* = 0.013), so we kept the time interaction. Also, the model with species interaction had smaller AIC (model 1: df = 10, AIC = 759 vs. no interaction: df = 6, AIC = 764) and there was a significant difference between both models (chi square = 12.8415, *p* = 0.013), so we kept the species interaction.

When nests were established (i.e., 24 h and 7 days after setup), there was a significant effect for the number of queens (z = 5.030, *p* < 0.001), species (z = 3.372, *p* < 0.001) and time for the workers to enter inside the nest (z = 4.776, *p* < 0.001) (Appendix A). These differences were mainly driven by the lower number of workers inside one-queen *L*. *humile* nests at 24 h after the setup than in each of the other treatments (Figure 3, Appendix A).

### 3.3. Workers Outside the Nest

The model “patrolling” without species interaction had a similar AIC than the model with the interactions (no interaction: df = 37, AIC = 248 vs. with interaction: df = 36, AIC = 249), and there was no significant difference between these models (Likelihood Ratio: LR = 0.042, *p* = 0.838), so we removed the species interaction. The model “nest maintenance” without species interaction had a similar AIC to the model with the interactions (no interaction: df = 4, AIC = 351, vs. with interaction: df = 5, AIC = 352), and there was no significant difference between these models (LR = 0.391, *p* = 0.532), so we removed the species interaction. The model “feeding” without species interaction had a similar AIC to the model with the interactions (no interaction: df = 4, AIC = 200 vs. with interaction: df = 5, AIC = 200) and there was no significant difference between these models (LR = 1.058, *p* = 0.304), so we removed the species interaction.

Once nests were established (i.e., 7 days after setup), we quantified the influence of the number of queens and species on the number of workers engaged in three outside nest tasks (i.e., patrolling, nest maintenance, and feeding). For both numbers of queens, *T. nigerrimum* nests had significantly fewer workers patrolling than *L. humile* nests did (z = −2.238, *p* = 0.025) (Figure 4a, Appendix A). The number of workers engaged in nest maintenance (z = −1.196, *p* = 0.232) and feeding (z = −0.830, *p* = 0.406) was similar for both species (Figure 4b,c, Appendix A). We found that six-queen nests had fewer workers engaged in nest maintenance (z = −2.980, *p* = 0.003) and feeding (z = −2.227, *p* = 0.026) compared to one-queen nests (Figure 4b,c, Appendix A).

## 4. Discussion

### 4.1. Queens during Nest Establishment

We found that queens belonging to nests with numerous queens entered the nest faster for both species. This result suggests that queens may influence the behavior of other queens during nest establishment. In polygynous species, it has already been demonstrated that queens may affect the reproductive behavior of other queens through pheromones that alter the oviposition rate [11,29]. To date, however, no study has determined whether specific queen pheromones influence the behavior of other queens during nest establishment.

Moreover, we did not find differences between species in time spent by the first queen to enter inside one-queen nests nor six-queen nests. This could be explained because both species have a similar colony-founding strategy. Indeed, the studied species typically have highly polygynous and polydomous colonies with a lack of aggressiveness that allow them to develop super colonies made up of interconnected nests [7].

### 4.2. Workers Inside the Nest

Our results suggest that for the time period analyzed from 0.5 h to 7 days after setup, the number of workers within the nest was higher in nests with a higher number of queens, especially for *T*. *nigerrimum*. Because the number of queens is related to the concentration of queen pheromones [12], nests with a higher number of queens could cause a high number of workers to follow them inside the nest [7,30]. Likely, the less time that workers spend outside the nest, the faster they will start to take care of the brood. Although ant colonies with more queens can grow faster, there might be a critical worker-to-queen ratio for colony growth and reproduction, as seen for some species such as *Mycocepurus smithii* [31]. For the invasive *L*. *humile*, an overabundance of queens promotes the seasonal execution of less productive queens [12]. According to our results, it seems that *T. nigerrimum* is more efficient than *L. humile* at establishing their nests, because the species depends less on the number of queens.

### 4.3. Workers Outside the Nest

*T*. *nigerrimum* nests had fewer workers engaged in patrolling compared to *L*. *humile* nests. Because our colonies were settled and maintained in the same conditions (i.e., humidity, temperature, light, food resources), the different numbers of workers engaged in patrolling could be explained by differences in the behavioral plasticity of species. Indeed, according to external and internal conditions, workers can shift between tasks to optimize colony performance [32]. Knowledge about the mechanisms underlying the behavioral plasticity of workers, such as genetic and developmental factors, would be the key to better understanding higher level processes such as species coexistence and competition between *T*. *nigerrimum* and *L*. *humile* colonies in invaded areas such as the Mediterranean basin.

For both species, we found that six-queen nests had fewer workers engaged in nest maintenance and feeding than one-queen nests. It is therefore likely that the number of workers engaged in those tasks could be modulated by the number of queens present in the nests. Indeed, a higher number of queens in the nests would require more workers for inside-nest tasks such as brood rearing and queen attendance rather than outside-nest tasks (i.e., nest maintenance and feeding). Other social characteristics, such as age, have been identified as one of the main factors promoting the task shifts of workers under stable conditions [30,32].

### 4.4. Perspectives

One of the main goals of invasion biology is predicting and preventing the future establishment of invasive species. Invasive species cause damage to the environment, and their impact and management have high economic costs [33]. Although a small fraction of introduced species become invasive [33,34,35], the continual increase in species trade and human-mediated exchange generates many new species introductions worldwide. Consequently, it is of high priority to better understand the processes, underlying causes, and consequences of establishment success. Among invasive species, more than 241 ant species have successfully colonized areas outside of their native range [36], and 19 of these species are listed in the Global Invasive Species Database (GISD).

According to the Convention on Biological Diversity [37], the most effective approaches to limit the impact of invasive species is to maintain and preserve diverse communities and habitats and prevent any introductions. Nevertheless, once introduced, there is a wide range of chemical, mechanical, and biological methods to limit the spread of invasive species [14,38]. Among invasive ant species, *L*. *humile* continues to spread [15] given the absence of an efficient biocontrol agent. In the Mediterranean basin, the presence of native species like *T*. *nigerrimum*, which has similar colony attributes and habitat requirements to *L*. *humile*, could be a promising approach to limit the spread of the *L*. *humile* super colony [18]. Thus, future studies should assess whether queen numbers can influence nest establishment success in situations in which both *L*. *humile* and *T*. *nigerrimum* are present.

## Figures and Tables

**Figure 1 animals-11-00591-f001:**
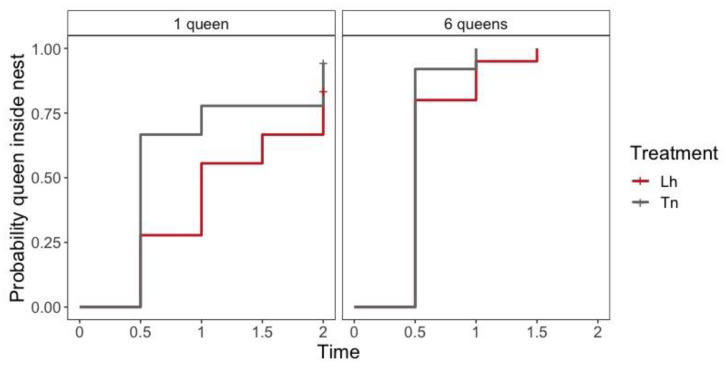
Kaplan–Meier plot of time (in hours) for first queen to enter the nest shows the survivor function stratified according to the number of queens and species. Plus signs (+) indicate censored data and their corresponding x values for the time at which censoring occurred. Lh = *L. humile* nest and Tn = *T. nigerrimum* nest.

**Figure 2 animals-11-00591-f002:**
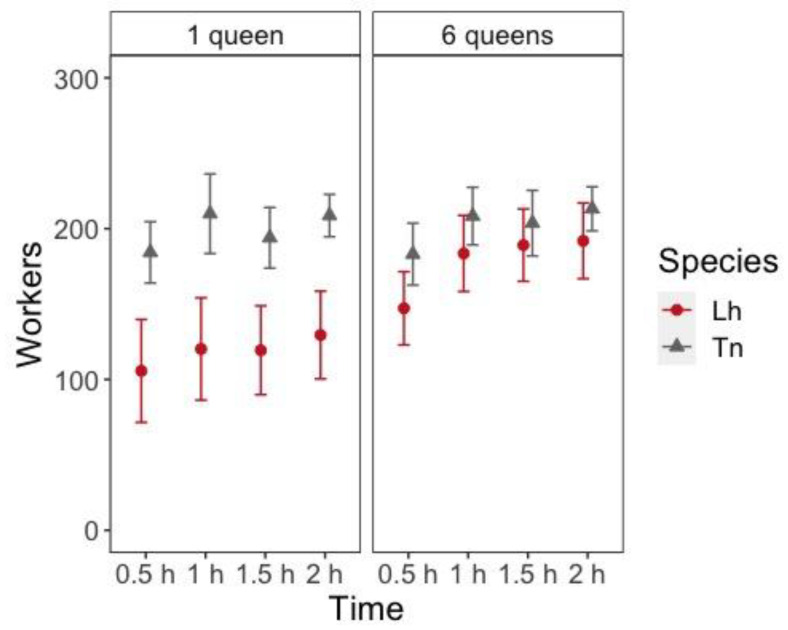
Number of workers (mean ± 95% Confidence Interval (CI)) inside the nest at different times after placing them in the foraging arena according to the species (Lh = *L. humile* and Tn = *T. nigerrimum*) and the number of queens in the colony.

**Figure 3 animals-11-00591-f003:**
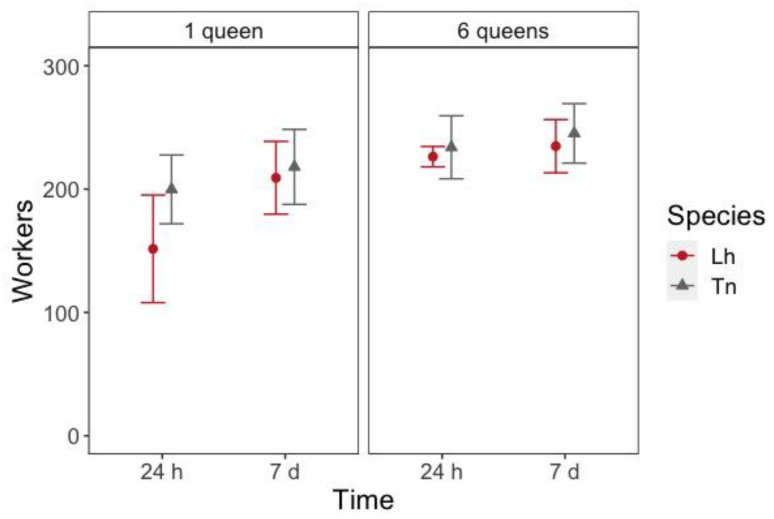
Number of workers (mean ± 95% Confidence Interval (CI)) inside the nest 24 h and 7 days after placing them in the foraging arena according to the species (Lh = *L. humile* and Tn = *T. nigerrimum*) and the number of queens in the colony.

**Figure 4 animals-11-00591-f004:**
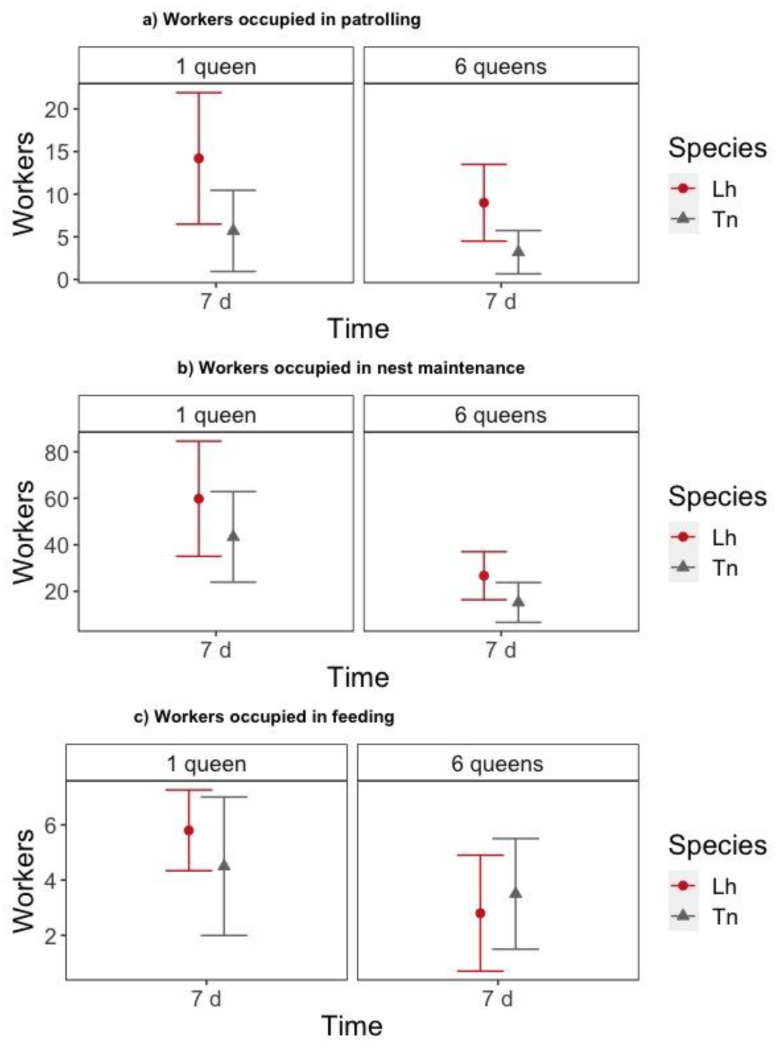
Number of workers (mean ± 95% Confidence Interval (CI)) outside the nest 7 days after placing them in the foraging arena according to the species (Lh = *L. humile* and Tn = *T. nigerrimum*) and the number of queens placed in the colony: (**a**) workers occupied in patrolling, (**b**) workers occupied in nest maintenance and (**c**) workers occupied in feeding.

## Data Availability

Data available on request.

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
