# Peer review of "Influence of the Number of Queens on Nest Establishment: Native and Invasive Ant Species"

_animals, 2021, doi:10.3390/ani11030591_

Round 1
Reviewer 1 Report
The authors carefully followed all my suggestions, I found the manuscript really improved. I believe it can be now suitable for publication.
Reviewer 2 Report
Dear authors,
thank you for the changes and clarifications in the new version of the manuscript. The revision has improved clarity and quality of the text, so I would recommend publication. I have noted some small grammatical errors in the PDF that may be corrected in he proofs.
Sincerely,
A. Laciny

This manuscript is a resubmission of an earlier submission. The following is a list of the peer review reports and author responses from that submission.
Round 1
Reviewer 1 Report
The study deals with the influence of the number of queens in nest establishment in two ant species, the invasive argentine ant Linepithema humile and the Mediterranean ant Tapinoma nigerrimum. The results highlight that Tapinoma is less dependent on the number of queens for the nest establishment. In my opinion, the most interesting point of the study is that the authors compare two species that are known to be possible antagonists, with Tapinoma acting as a fence for the spreading of the Linepithema. I found particularly interesting the result that the activity outside the nest is higher in Linepithema than Tapinoma even with many queens inside the nest, suggesting a more intense activity that might imply higher competitiveness in foraging or resource finding. Overall, the results of the study are sufficiently supported by the experimental design, the manuscript is sufficiently clear in the aims and it is, in general, well-written. However, I think that the part of the statistical analyses should be a little bit straightened. There is probably too much information, and it might result confused for the readers. Sometimes probably the dataset should be split. These are my suggestions (feel free not to follow all of them):
- KM survival curves to assess the speed of queens entering the nest. I believe that the data are too much unbalanced between colonies with 1 and 6 queens (120 vs. 18 values), thus it is quite hard to compare the curves. You may consider splitting the dataset and separately compare the 1 queen colonies and the 6 queen colonies between the two species. Otherwise, you might consider to include in the analysis the time elapsing before only the first queen entered the nest, not all the 6 queens. It could be an interesting datum anyway. In fact, the subsequent queens could be influenced by the presence of queens already entered within the nest (seek studies dealing with this matter). After using the log-rank test to assess the differences among curves (you may alternatively consider using Cox models), you could perform multiple comparisons to verify the differences in pairs and report the result in the main text.
- In the GLMM and GLM models to assess the relationship between the number of workers and the other factors, you always tested the most complex model, namely the one with all factors added with their interaction. I think that you need to verify that this model is actually the best one by ranking models (using the AIC index) made by single variables, couple of variables and their linear sum without interaction, and so on, with a stepwise approach. In this way, you can focus only on those variables having a significant weight on the dependent variable, and then you can test them. As you did, is quite hard to understand which variable is really significant. Moreover, you used time as a factor, but I’m not sure it is the best way to manage this variable. You could try to use it as a continuous variable, though you measured the variables only in those 4 steps.
- In the tables, do not report the name of only one level (i.e. 6-queen, Tn, etc.), but write the name of the variable (time, species, queens). The statistic for the intercept can be omitted.
- In lines from 187 to 214 are reported several p values in parentheses, but none of them is preceded by its statistical values. Please add them.
- In my opinion, except for the first image showing the experimental apparatus, the rest of the supplementary material is uninformative and can be removed.
Other minor comments:
Line 72: Please write the complete name of the species, is the first time you mention it after the abstract.
Lines 262-264: Here you should probably add some references for these statements.
Author Response
"Please see the attachment."

Reviewer 2 Report
Castaneda et al.
I will start with the specific points.
Lines 192-194. The authors claim that the number of queens in Tn had a significant effect on the number of workers in the nest. However, looking at the data plotted on the associated Figure 2, I cannot see how this could possibly be true. The plotted values look almost identical.
Lines 206-214. The effects are again claimed to be significantly negative, when the figures show a positive effect – worker number is increasing over time.
Across Tables 1-3, the results for Lh are not marked. Only Tn is listed.
In figure and table captions, unless it is journal policy, it is usual to abbreviate the genus; e.g., T. and L.
General concerns.
Couldn’t your results be entirely explained if Tn simply moves or runs faster than Lh as a general rule? Did you measure the velocity at which workers and queens normally moved during your experiments?
The Introduction and Discussion placed the work in the context of understanding why some species are invasive and others are not, and how to limit the spread of invasives. However, I do not see any connection between the results presented here and those wider issues. At best, you are comparing two species – and one might always expect there to be differences based on that alone. What about the differences that would make Lh and not Tn to be better at invading?
After all, Lh likely evolved its queen/worker distribution patterns when it was a native in its native habitat. Now if you compared the behavior in an invasive population to the behavior of the original native population, and found a difference, then you might have evidence for selection for “invasive” traits.
Respectfully submitted,
Author Response
"Please see the attachment."

Reviewer 3 Report
This study by Castañeda et al. provides new insights into the nest dynamics of of native and invasive species of ants. While the topic is of interest, some key issues should be addressed to make it suitable for publication (see also comments to PDF):
L12: native to which region?
L60: please provide a reference that shows the relevance of queen number on nest establishment.
L71-72: As ecological factors in the natural world are certainly crucial for the establishment of invasive species, please explain in this section a) the ecological requirements and life history traits of the two studied species (L298-303 may be moved here), and b) how and why it is suitable to conduct this study purely in a lab setting.
L94: Does this temperature range represent natural conditions / species requirements?
L101: Why where exactly 1 or 6 queens chosen?
L105: correct citation format; were the same species studied here?
L112: Which photo equipment was used?
L129-130: Where does this definition via the 70% threshold come from, please provide a reference.
L139: Why was GLMM chosen over other statistical methods (e.g. PCA)? As it is, the resulting Tables are quite hard and not intuitive to interpret.
L158: Why is patrolling defined as walking on the wall vs. the floor?
L183-184: Reference to the supplementary table should be omitted from figure legend.
L188-194: This section seems problematic, as it ascribes direct causality (queen presence “increased“ number of workers, see also L260) and the conclusions that are drawn do not seem to be supported by the graphical representation in Fig. 2 – please explain in more detail how these significant results have come to be. Same with L210 and Fig. 3.
Tables: There seems to be something wrong with the display of significance codes (0 ‘***’ 0.001 ‘**’ 0.01 ‘*’ 0.05 ‘.’ 0.1 ‘ ‘ 1 )
L257: Please avoid anthropomorphic language, ants are probably not “concerned“
L258: This is interesting, please give more detail / references about risk avoidance.
L264: Provide reference for the connection between worker behaviour and diploid eggs.
Supplements legends: “rangs“ - do you mean rows? “show and homoscedasticity“ - is there a word missing?

Author Response
"Please see the attachment."
